# Optimizing Pancreatic Cancer Therapy: The Promise of Immune Stimulatory Oncolytic Viruses

**DOI:** 10.3390/ijms25189912

**Published:** 2024-09-13

**Authors:** Shivani Thoidingjam, Aseem Rai Bhatnagar, Sushmitha Sriramulu, Farzan Siddiqui, Shyam Nyati

**Affiliations:** 1Department of Radiation Oncology, Henry Ford Health, Detroit, MI 48202, USA; 2Henry Ford Health + Michigan State University Health Sciences, Detroit, MI 48202, USA; 3Department of Medicine, Michigan State University, East Lansing, MI 48824, USA; 4Department of Radiology, Michigan State University, East Lansing, MI 48824, USA

**Keywords:** oncolytic viruses (OVs), cancer immunotherapy, DNA and RNA viruses, advanced pancreatic adenocarcinoma

## Abstract

Pancreatic cancer presents formidable challenges due to rapid progression and resistance to conventional treatments. Oncolytic viruses (OVs) selectively infect cancer cells and cause cancer cells to lyse, releasing molecules that can be identified by the host’s immune system. Moreover, OV can carry immune-stimulatory payloads such as interleukin-12, which when delivered locally can enhance immune system-mediated tumor killing. OVs are very well tolerated by cancer patients due to their ability to selectively target tumors without affecting surrounding normal tissues. OVs have recently been combined with other therapies, including chemotherapy and immunotherapy, to improve clinical outcomes. Several OVs including adenovirus, herpes simplex viruses (HSVs), vaccinia virus, parvovirus, reovirus, and measles virus have been evaluated in preclinical and clinical settings for the treatment of pancreatic cancer. We evaluated the safety and tolerability of a replication-competent oncolytic adenoviral vector carrying two suicide genes (thymidine kinase, TK; and cytosine deaminase, CD) and human interleukin-12 (hIL12) in metastatic pancreatic cancer patients in a phase 1 trial. This vector was found to be safe and well-tolerated at the highest doses tested without causing any significant adverse events (SAEs). Moreover, long-term follow-up studies indicated an increase in the overall survival (OS) in subjects receiving the highest dose of the OV. Our encouraging long-term survival data provide hope for patients with advanced pancreatic cancer, a disease that has not seen a meaningful increase in OS in the last five decades. In this review article, we highlight several preclinical and clinical studies and discuss future directions for optimizing OV therapy in pancreatic cancer. We envision OV-based gene therapy to be a game changer in the near future with the advent of newer generation OVs that have higher specificity and selectivity combined with personalized treatment plans developed under AI guidance.

## 1. Introduction

Pancreatic cancer remains one of the most lethal malignancies, with a five-year survival rate of 13% [1]. The primary reasons for this dismal prognosis include late diagnosis, aggressive tumor biology, and limited treatment options [2]. Traditional therapies such as surgery, chemotherapy, and radiation often fall short due to the tumor’s resistance mechanisms and the complex tumor microenvironment (TME) [3]. While complete surgical resection can significantly extend patient survival and is considered the only curative approach, only about 20% of patients are eligible for surgery, mainly due to diagnosis at an advanced stage [4,5,6]. The standard-of-care for metastatic pancreatic adenocarcinoma is continuous chemotherapy with FOLFIRINOX or gemcitabine and nanoparticle albumin-bound paclitaxel based on the PRODIGE-4/ACCORD-11 [7] and MPACT trials [8]. While initial stages of chemotherapy often lead to patient improvement, most eventually develop drug resistance, which severely impacts their prognosis [9,10,11,12]. Consequently, there is an urgent need for innovative therapeutic strategies that can overcome these barriers and improve patient outcomes.

The potential of viruses as anticancer agents was first recognized in the early 20th century when physicians observed tumor regression in some cancer patients following viral infections. These initial findings sparked the idea of using viruses to fight cancer. By the 1960s, research into oncolytic viruses gained momentum, with scientists investigating naturally occurring viruses for their cancer-killing properties. However, early studies faced challenges such as uncontrolled infections and immune responses that limited viral efficacy. The field was revolutionized in the late 20th century with advances in genetic engineering, which led to the development of viruses that selectively target cancer cells while minimizing their pathogenicity [13,14]. In recent years, immunotherapy has revolutionized cancer treatment, offering new hope where conventional therapies have failed [15]. Among emerging modalities, OVs have garnered significant attention for their dual ability to selectively infect and lyse cancer cells while simultaneously stimulating robust antitumor immune responses to destroy neighboring cancer cells [16]. OVs are either naturally occurring oncotropic viruses or genetically engineered to selectively infect and destroy cancer cells while sparing normal tissues [17]. OVs exploit the unique characteristics of cancer cells, such as abnormal responses to stress, impaired antiviral responses, and specific surface receptors, to selectively infect them. For example, reovirus targets cells with Ras mutations, while vesicular stomatitis virus (VSV) and Newcastle disease virus preferentially infect cells with interferon response-related mutations [16,17]. Certain viruses utilize specific cell surface receptors; poliovirus targets CD155, while measles virus targets CD46 or CD150 [18]. Other viruses like herpes simplex virus type 1 (HSV-1) take advantage of malignancy-driven cellular changes, including alterations in the extracellular matrix for infecting cancer cells [19].

### 1.1. Oncolytic Viruses and Mechanism of Action

OVs employ a multifaceted mechanism of action that includes both direct tumor cell destruction (oncolysis) and the stimulation of the immune system. Upon infection, the virus replicates within the cancer cell and hijacks the cell’s protein production machinery leading to cell lysis and the release of new viral particles that can infect adjacent cancer cells, further propagating the oncolytic effect [20]. The lysis of tumor cells releases tumor antigens, damage-associated molecular patterns (DAMPs), and pathogen-associated molecular patterns (PAMPs). These components trigger host immune responses and activate antitumor immunity, which further activates T cells and other immune responses against tumors [21,22]. The presence of viral particles and tumor antigens thus stimulate both the innate and adaptive immune responses and transform the immunosuppressive TME to a more immunologically responsive state [23,24]. The modulation of the TME by OVs is further discussed in detail later in this review.

Additionally, OVs with natural infection abilities can be genetically modified to improve their immunogenicity and effectiveness against cancer cells [18]. This enhancement involves knocking out certain genes to reduce infection of normal cells, inserting new genes to boost oncolytic activity, and transferring foreign genes to improve immune responses [25]. Tumor-specific promoters ensure the virus replicates selectively in cancer cells. Engineered OVs can express cytokines such as TNF, GM-CSF, IL-7, IL-12, and IFN-β, which enhance cell-lysing capabilities and stimulate antitumor immunity [26,27,28,29]. These modifications improve safety, tumor specificity, and potency while reducing pathogenicity. Modified OVs evaluated in clinical trials include adenoviruses, vesicular stomatitis, vaccinia, measles, and herpes simplex viruses [17].

Furthermore, OVs can be combined with other cancer therapies such as chemotherapy, radiation therapy, or immunotherapy to enhance their effectiveness [30]. For example, combining OVs with agents that target the dense stromal environment of tumors like pancreatic tumors can improve viral delivery and penetration into the tumor mass [31]. Overall, OVs may offer several advantages over conventional immunotherapies, including accurate targeting, high efficacy in killing cancer cells, and minimal side effects [32,33,34]. This makes OVs particularly suited for overcoming significant challenges posed by pancreatic cancer.

### 1.2. Types of Oncolytic Viruses

OVs are classified based on their genetic material into DNA and RNA viruses. OVs with DNA as genetic material include adenoviruses, herpes simplex virus (HSV), parvoviruses, and poxviruses (e.g., vaccinia virus, myxoma virus). OVs based on RNA viruses include coxsackievirus, maraba virus, measles virus (MV), Newcastle disease virus (NDV), poliovirus, reovirus, retroviruses, Seneca Valley virus (SVV), Semliki Forest virus (SFV), vesicular stomatitis virus (VSV), and sindbis virus (SBV). Each virus type has unique characteristics, advantages, and limitations, making them suitable for different cancers and treatment strategies [34].

### 1.3. Oncolytic Viruses and the Tumor Microenvironment

Pancreatic cancer is characterized by a challenging tumor microenvironment and dense, fibrotic tissue known as desmoplasia [35]. This desmoplastic stroma, composed of both cellular and noncellular components, forms a thick extracellular matrix (ECM) barrier around the tumor [35] that hinders the effectiveness of treatments and promotes tumor growth and invasion [36]. Moreover, pancreatic cancer is thought to be an “immune cold” tumor, with an immunosuppressive TME [37] that helps the tumor evade the body’s natural defenses and supports cancer progression [38].

OVs can alter the TME to boost immune activation and destroy cancer cells in several ways [39]. First, they exploit cancer cells’ dysfunctional interferon (IFN) pathway for easy infection [40]. OVs induce immunogenic cell death (ICD), releasing molecules like calreticulin, ATP, and HMGB1 [41], which activate dendritic cells (DCs) and T lymphocytes, overcoming the tumor’s immune evasion [21]. This local immune activation often leads to a long-lasting anticancer response, even in advanced stages [42].

Additionally, OVs influence tumor vasculature, tumor-associated fibroblasts, ECM, and various other components [43]. Oncolytic viruses (OVs) can target both nascent and established blood vessels in tumors without harming normal vasculature. They can directly infect and lyse tumor endothelial cells, induce immune responses that reduce tumor perfusion, and express viral proteins with antiangiogenic properties [44]. Genetically engineered OVs can additionally deliver antivascular agents to further enhance these effects [45].

Cancer-associated fibroblasts (CAFs) and the ECM are crucial components of the TME that support tumor growth and metastasis [38]. OVs can disrupt the interaction between cancer cells and CAFs leading to a less supportive environment for tumor growth. Ilkow et al. demonstrated that vesicular stomatitis virus (VSV)-based therapeutics are enhanced through interactions between CAFs and cancer cells [46]. Transforming growth factor-beta 1 (TGF-β1) secreted by tumor cells promotes OV infection in CAFs, while high levels of fibroblast growth factor 2 (FGF2) make tumor cells more susceptible to viral infection [46]. Additionally, oncolytic adenoviruses (OAds) have been shown to target both glioblastoma cells and glioblastoma-associated stromal FAP+ cells, effectively disrupting tumor and stromal cell interactions [45].

To enhance their spread within tumors, OVs use strategies like expressing ECM-degrading proteins such as relaxin and decorin [47,48,49]. Hyaluronidase-armed adenoviruses such as VCN-01 demonstrated promise in overcoming ECM barriers in clinical trials, including in pancreatic cancer (NCT02045589, NCT02045602) [50,51]. Additionally, OVs can convert unresponsive “cold” tumors into more responsive “hot” tumors, thus enhancing the effectiveness of immunotherapy [23,24,52]. Thus, OVs not only directly kill cancer cells but also reshape the TME to enhance immune-mediated destruction of tumors. By inducing inflammation, normalizing vasculature, disrupting stromal components, targeting the ECM and the crosstalk between the TME components, OVs create an environment hostile to cancer cells and improve clinical outcome. The mechanisms of OV activity and tumor microenvironment remodeling in pancreatic cancer are shown in Figure 1.

### 1.4. Development of OVs

OVs as a cancer treatment strategy have evolved significantly over the years. The concept of using viruses for cancer therapy dates to the late 19th century [53], but initial trials with wild-type RNA viruses and adenoviruses in the mid-20th century had limited success [54]. Interest in OVs as anticancer agents increased in the 1990s with the demonstration of genetically engineered herpes simplex virus (HSV) [55]. Since then, a diverse range of DNA and RNA viruses tailored for tumor specificity and safety have entered clinical trials [54]. Key breakthroughs include FDA and EMA approval of Talimogene Laherparepvec (T-Vec; Herpes simplex virus) in 2015 for advanced melanoma in the USA and Europe [56]. Other country-specific approved OVs include Rigvir (picornavirus, treatment of melanoma, Latvia), H101 (adenovirus, treatment of head and neck cancer, China), and DELYTACT (Herpes simplex virus, treatment of brain cancers such as glioblastoma, Japan) [57].

Today, OVs represent a promising frontier in cancer therapy, with ongoing research exploring their efficacy across various cancer types and refining treatment approaches. Numerous preclinical and clinical studies have demonstrated the safety and efficacy of OVs in various cancer models, including glioblastoma, breast, prostate, and pancreatic cancers [17,58,59,60].

## 2. Oncolytic Viruses for Pancreatic Cancer

### 2.1. Preclinical Studies of Oncolytic Viruses in Pancreatic Cancer

Precise tumor targeting is crucial for the effective systemic delivery of OVs. Several different viruses including adenovirus, HSV, vaccinia virus, parvovirus, reovirus, and measles virus have been tested and validated in preclinical pancreatic cancer models. Some key studies are discussed below.

Delta-24-RGD (DNX-2401) is an OV designed to selectively replicate in tumor cells with p16/RB/E2F pathway abnormalities. It is an OAd with a 24-base pair deletion in the E1A region and an RGD-4C modification in the virus fiber which enhances its ability to infect cancer cells independently of coxsackievirus and adenovirus receptor expression. DNX-2401 significantly reduced tumor growth and enhanced anticancer activity in pancreatic cancer models, particularly when combined with the phosphatidylserine-targeting antibody 1N11, suggesting synergistic anticancer immune responses [61]. Similarly, a neurotensin peptide-conjugated polyethylene glycol (PEG-NT)-coated OAd was engineered to express both decorin (DCN) and a soluble Wnt decoy peptide sLRP6E1E2 (oAd/DCN/LRP-PEG-NT) specifically targeting pancreatic cancer [62]. This dual-function approach aims to degrade the extracellular matrix (ECM) and disrupt Wnt signaling, thus enhancing the therapeutic efficacy in pancreatic tumors. OAd/DCN/LRP-PEG-NT effectively targeted neurotensin receptor 1 (NTR)-overexpressing pancreatic cancer cells. It increased cancer cell killing, improved transduction efficiency, reduced immune responses, prolonged blood retention, and significantly suppressed tumor growth in vivo [62]. Na et al. explored a novel approach to improve systemic delivery of OAds using human bone marrow-derived mesenchymal stromal cells (hMSCs) as carriers. hMSCs were chosen due to their natural ability to migrate to tumors and their low immunogenicity, which protects the virus from the immune system. OAd was complexed with relaxin to help break down the dense ECM, allowing better penetration of the virus, and biodegradable polymer (PCDP) to boost the internalization of the virus into hMSCs [63]. The oAd/RLX-PCDP complex demonstrated enhanced internalization, superior viral replication and release within tumors, and relaxin expression, resulting in a stronger antitumor effect compared to naked oAd/RLX or oAd/RLX-treated hMSCs. The study suggests that hMSCs, along with relaxin-expressing oAd, could effectively overcome the barriers of the tumor microenvironment, offering a promising strategy for improving pancreatic cancer treatment [63]. Another study targeting the desmoplastic TME of pancreatic cancer investigated the effect of ECM-degrading relaxin expressing OAd (YDC002) combined with gemcitabine in chemo-resistant pancreatic cancer. YDC002 degraded ECM, overcame ECM-induced chemoresistance, and enhanced gemcitabine-mediated cytotoxicity and oncolytic effects [47]. Brugada-Vilà et al. aimed to enhance the systemic delivery and efficacy of oncolytic adenoviruses by coating them with PEGylated oligopeptide-modified poly(β-amino ester)s (OM-pBAEs). This formulation improved transduction, evaded neutralizing antibodies, reduced liver sequestration, and enhanced therapeutic potential in PDAC mouse models compared to noncoated viruses [64].

Similarly, IFN-α expressed from oncolytic adenoviruses synergistically increased the effectiveness of radiation and chemotherapy (5-FU, gemcitabine, and cisplatin) for in vitro and in vivo pancreatic cancer models. This study highlighted that combining IFN-expressing oncolytic adenoviruses with chemoradiation offered a promising innovative approach for pancreatic cancer patients, particularly those unable to tolerate standard chemotherapy [65]. Watanabe et al. demonstrated that combining mesothelin-redirected chimeric antigen receptor T cells (meso-CAR T cells) with an oncolytic adenovirus expressing TNF-α and IL-2 (OAd-TNFa-IL2) significantly enhanced antitumor efficacy in pancreatic ductal adenocarcinoma (PDAC) models. This approach increased tumor-infiltrating lymphocytes (TILs), enhanced T-cell function, prevented tumor metastasis, and induced tumor regression [66]. OAds are designed to target cancer cells, but genetic variability in tumors, like abnormal miRNA expression in pancreatic cancer, can hinder their replication. Raimondi et al. found that inhibiting the overexpressed miR-222 in cancer cells sensitized them to viral oncolysis. A novel OAd engineered to reduce miR-222 (AdNuPARmE1A-miR222-S) enhanced viral replication and cytotoxicity and effectively controlled tumor growth in vivo. The improved antitumor effects resulted from miR-222 inhibition and restoration of target [67].

To address the challenges faced by traditional Ad5-based OAds in entry into cancer cells due to low levels of coxsackievirus and adenovirus receptor (CAR), Doerner et al. developed novel (OAds) derived from different serotypes [68]. These new OAds include a tumor-selective mutation to allow the viruses to selectively replicate in tumor cells and express the RNAi inhibitor P19 to enhance viral replication and oncolytic activity. OAds based on Ad1, Ad2, and Ad6 were more effective at lysing cancer cells than the traditional Ad5-based OAd [68]. Hashimoto et al. developed a telomerase-specific OAd armed with the p53 gene (OBP-702) and investigated its potential to promote long-term antitumor immunity. OBP-702 increased effector memory precursor cells and led to tissue-resident and effector memory T cells in murine pancreatic tumors. OBP-702 in combination with gemcitabine and nab-paclitaxel (GN) maintained memory T-cell activation and showed significant antitumor effects. In a neoadjuvant model, GN with OBP-702 provided long-term antitumor effects post-tumor resection, highlighting OBP-702’s potential as a long-term immunostimulant in pancreatic cancer [69]. Ge et al. [70] developed an OAd that expressed tumor necrosis factor-related apoptosis-inducing ligand (TRAIL) and second mitochondria-derived activator of caspase (Smac) (ZD55-TRAIL-IETD-Smac) and evaluated its activity in pancreatic cancer models with a CDK inhibitor SNS-032. SNS-032 improved virus-induced apoptosis and cell death through modulating antiapoptotic signaling pathways and significantly reduced tumor growth in vivo [70].

The propensity of reoviruses to replicate and induce cell death specifically in cells with activated Ras led to the development of Reolysin (Pelareorep). Reolysin, a reovirus serotype-3-Dearing strain targets tumors with Ras pathway mutations which are present in up to 70% of pancreatic cancers. Carew et al. found that Reolysin boosts selective reovirus replication and reduces cell viability in KRas-transformed human pancreatic cells and pancreatic cancer cell lines [71]. Additionally, combining Reolysin with ER stress inducers tunicamycin and bortezomib intensified its anticancer effects in both in vitro and in vivo models [71]. Similarly, Baertsch et al. integrated synthetic microRNA target sites (miRTS) into the genome of oncolytic measles viruses (MVs) virus to enhance selectivity and safety [72]. They developed MV-EGFPmtd with miRTS for miR-122, miR-7, and miR-148a, which are abundant in vital organs. V-EGFPmtd’s replication was inhibited in cell lines and primary hepatocytes expressing these microRNAs but maintained oncolytic potency in pancreatic cancer models. This observation demonstrates the feasibility of targeting multiple microRNAs to modify oncolytic vector tropism without compromising efficacy [72]. Likewise, tumor-targeted oncolytic vaccinia virus (VV) with deleted thymidine kinase armed with murine IL-10 (VVLΔTK-IL-10) was developed. VVLΔTK-IL-10 increased survival in immunocompetent mice [73]. Wang et al. developed an oncolytic herpes simplex virus-1 (oHSV) armed with CD40 ligand (oHSV-CD40L) to potentially enhance the antitumoral immune response in PDAC. CD40L activates immune cells, particularly dendritic cells and T cells, by binding to the CD40 receptor. Intratumoral administration of oHSV-CD40L effectively slowed tumor growth, prolonged survival, increased mature dendritic cells, activated cytotoxic T cells, and reduced regulatory T cells [74]. In an immunocompetent syngeneic PDAC model, oHSV treatment reduced tumor burden and improved survival. Single-cell RNA sequencing and FACS analysis revealed that oHSV decreased tumor-associated macrophages and boosted tumor-infiltrating lymphocytes, including activated CD8+ T and Th1 cells, leading to enhanced PDAC responsiveness to immunotherapy [24].

Vienne et al. explored the oncolytic potential of the fibrotropic minute virus of mice prototype (MVMp) in PDAC [75]. MVMp selectively infected, replicated in, and killed PDAC cells while sparing epithelial-type cells. In immune-competent mouse models, MVMp not only inhibited tumor growth and extended survival but also enhanced immune cell infiltration into tumors, particularly myeloid cells and cytotoxic T cells, with a less exhausted phenotype. The study identified a five-protein classifier (FAK, N-cadherin, E-cadherin, β-catenin, Snail) that predicted a response to MVMp which could guide patient selection for virotherapy [75]. Schäfer et al. evaluated the effectiveness of OVs against PDAC by testing them on fourteen patient-derived PDAC cultures, reflecting the disease’s heterogeneity. Twelve of the fourteen cultures responded to at least one OV, but no single virus was universally superior. Sensitivity to OVs varied by PDAC subtype, with the quasi-mesenchymal/basal-like subtype being more responsive to certain OVs like H-1PV, jin-3, and T-VEC. Key findings included Galectin-1 as a potential biomarker for H-1PV effectiveness and high interferon-stimulated gene (ISG) expression as a marker of resistance to the oncolytic measles virus. Combining the measles virus with a cGAS inhibitor enhanced tumor cell killing. The study highlights the need for personalized treatment strategies due to PDAC’s heterogeneity [76]. These findings collectively illustrate the diverse strategies and promising outcomes of oncolytic viruses in preclinical models of pancreatic cancer. Table 1 presents a non-exhaustive list of preclinical studies based on oncolytic viruses (OVs) in pancreatic cancer.

### 2.2. Preclinical Trials That Combine OV with Immunotherapy

Programmed cell death-1 (PD-1), programmed cell death-ligand 1 (PD-L1), and cytotoxic T lymphocyte-associated antigen-4 (CTLA-4) are key in regulating immune responses to prevent autoimmunity. Immune checkpoint inhibitors (ICIs) that block these receptors enhance T-cell responses against tumors. OVs not only cause direct tumor cell lysis but also stimulate antitumor immune responses, making them excellent candidates for combination with ICIs and adoptive cell therapy (ACT). OVs can induce ICD, releasing tumor antigens and damage-associated molecular patterns (DAMPs) that promote dendritic cell maturation and T-cell activation. This synergistic combination can potentially overcome monotherapy limitations and improve therapeutic outcomes [86,87]. Recently, numerous research groups have explored the combined use of OV therapy and immunotherapy to treat a variety of cancers, including pancreatic cancers. In this section, we discuss several studies that explored different OVs in combination with immunotherapy or immunomodulatory therapies.

Liu et al. developed an HSV1 OV expressing murine OX40 ligand (OV-mOX40L) [88]. OX40 ligand stimulates T-cell proliferation and activation. OV-mOX40L increased tumor-infiltrating CD4+ T cells, reduced CTL exhaustion and regulatory T cells, reprogrammed pro-inflammatory macrophages and neutrophils, and decreased CAF in mice. Combination of OV-mOX40L with antiIL6 (BE0046) and antiPD-1 (BE0146) agents significantly prolonged survival in PDAC mice, suggesting a promising multimodal therapeutic strategy for PDAC [88].

Similarly, NKT cell activation therapy with an oncolytic vesicular stomatitis virus (VSVΔM51) engineered to express IL-15 (VSV-IL-15) was evaluated in a mouse PDAC model [89]. Here, C57BL/6 mice bearing Panc02 tumors were treated with either VSV-GFP or VSV-IL-15 along with α-GalCer-loaded dendritic cells for NKT cell activation. A group of mice additionally received antiPD-1 antibodies after NKT cell activation. The combination treatment led to better tumor control, validating the superiority of this approach for pancreatic cancer treatment [89].

### 2.3. OV-Based Clinical Studies in Pancreatic Cancer

The safety, feasibility, and clinical activity of several oncolytic OVs have been evaluated in phase 1 or 2 trials as single agents and in combination with chemotherapy and immunotherapy in patients with pancreatic cancer. The majority of these trials utilized modified oncolytic adenoviruses, while one trial used oncolytic reoviruses. Several selected clinical studies are discussed here.

ONYX-015 (dl1520) adenovirus: ONYX-015 is an adenovirus with a deletion of 55 kDa in the E1B gene that enables selective replication and increased lysis of cancer cells that carry p53 mutations. ONYX-015 (dl1520) was administered directly in primary tumors in patients with unresectable pancreatic cancer in a phase-I (NCT00006106) dose-escalation trial [90]. Multiple doses of the virus were very well tolerated by study subjects up to the maximum doses tested (2 × 10^12^ particles) [90]. A follow-up phase-I/II trial (NCT00006106) assessed the feasibility, tolerability, and efficacy of ONYX-015 with gemcitabine in 21 patients with unresectable/metastatic pancreatic cancer [91]. Treatment involved eight sessions of ONYX-015 injected via endoscopic ultrasound (EUS) into the primary tumor with the final four sessions in combination with gemcitabine. Overall, the study highlighted the feasibility of transgastric EUS-guided delivery for biological agents and the manageable safety profile of ONYX-015 [91]. Although ONYX-015 showed limited efficacy, these trials demonstrated the feasibility, tolerance, and safety of adenoviral delivery. This success laid the groundwork for future clinical evaluations with more advanced OVs [92].

VCN-01 adenovirus: A phase I trial (NCT02045602) evaluated the safety, maximum tolerated dose (MTD), and recommended phase II dose of VCN-01 for advanced pancreatic adenocarcinoma [58]. VCN-01 is an OAd designed to replicate in cancer cells with defective RB1 pathways and express hyaluronidase. Patients received escalating doses of VCN-01 combined with nab-paclitaxel and gemcitabine. The trial found no dose-limiting toxicities and achieved a 50% overall response rate. The study concluded that VCN-01, in combination with chemotherapy, is feasible, safe, and promising for treating pancreatic adenocarcinoma [93].

Ad5-yCD/mutTKSR39rep-hIL-12 adenovirus: We conducted a phase I trial [93] for metastatic pancreatic cancer using a replication-competent adenovirus Ad5-yCD/mutTKSR39rep-hIL12 that expresses two suicide genes (yeast cytosine deaminase/mutant S39R thymidine kinase from HSV) and human interleukin-12 (IL-12) (NCT03281382). The trial’s primary endpoints were the MTD and dose-limiting toxicities up to day 21. Most of the adverse events were mild, and MTD was not reached. The presence of Ad5-vector DNA indicated viral replication, while an increase in serum levels of IL-12, IFN-γ, and CXCL10 confirmed immune activation/potentiation [93]. Long-term follow-up showed that patients in the highest dose cohort had a higher median overall survival (18.4 months) compared to 4.8 months and 3.5 months in the lower dose cohorts [58].

LOAd703 adenovirus: LOAd703 encodes immunostimulatory genes TMZ-CD40L and 4-1BBL. A nonrandomized phase I/II trial (NCT02705196) assessed the safety and feasibility of LOAd703 with chemotherapy for advanced PDAC [94]. Twenty-one patients received standard chemotherapy (nab-paclitaxel and gemcitabine) and intratumoral LOAd703 injections. The treatment was generally well-tolerated, with common side effects being fever, fatigue, and elevated liver enzymes. No MTD was identified, confirming that the highest dose evaluated was safe. Immunological responses included increased adenovirus-specific T cells and CD8+ effector memory cells. About half (44%) of evaluable patients showed objective responses [94]. Arm 2 of this trial (LOKON001), which includes atezolizumab in addition to nab-paclitaxel plus gemcitabine and LOAd703, is ongoing.

Pelareorep (Reolysin) reovirus: A randomized phase II study (NCT01280058) evaluated pelareorep, an oncolytic reovirus, in combination with carboplatin and paclitaxel in treatment-naive metastatic pancreatic adenocarcinoma patients [95]. Pelareorep was found to be safe; however, it did not improve PFS when combined with carboplatin/paclitaxel (compared to chemo alone). Further targeting of immunosuppressive mediators may enhance oncolytic virotherapy [95].

Recombinant Human Adenovirus Type 5 (H101): Malignant ascites, a complication from peritoneal spread of malignancies, lacks effective treatments. In a phase II trial (NCT04771676), oncolytic adenovirus H101 virotherapy extended the median time to repeat paracentesis to 45 days, showed tolerable toxicity, and enhanced CD8+ T-cell and macrophage immune responses, suggesting the potential for combination with antiPD(L)1 therapy [96].

### 2.4. Clinical Studies Combining Oncolytic Viruses with Immunotherapy in Pancreatic Cancer

Reolysin was combined with chemotherapy and ICI (pembrolizumab) in patients with relapsed metastatic adenocarcinoma of the pancreas (NCT00998322) in an open-label phase 1b study [97]. Eleven patients received Reolysin, pembrolizumab, and either 5-FU/LV, gemcitabine, or irinotecan, every 3 weeks until disease progression or unacceptable toxicity. The primary endpoint was safety, with secondary objectives including tumor response and immune analysis. Grade 3 or 4 adverse events occurred in 73% of patients, including abdominal pain, anemia, biliary obstruction, and neutropenia. Among five evaluable patients, one achieved a partial response, and two had stable disease. On-treatment biopsies showed reovirus infection in cancer cells and immune infiltrates. Seven patients died due to disease progression [97].

A phase I study from the same group (NCT02620423) evaluated pelareorep (Reolysin) combined with pembrolizumab and chemotherapy in patients with advanced pancreatic cancer [98]. The study aimed to determine safety and efficacy in patients who had progressed after first-line treatment. Eleven patients received pelareorep, pembrolizumab, and either 5-fluorouracil, gemcitabine, or irinotecan. The treatment was well-tolerated, with most adverse events being mild flu-like symptoms. Disease control was achieved in three patients, out of which one had a partial response and two had stable disease. On-treatment biopsies showed reovirus replication, and T-cell receptor sequencing revealed the emergence of new T-cell clones. Patients with clinical benefit showed high peripheral clonality and changes in immune gene expression [98].

MEM-288 is an oncolytic adenovirus vector engineered to selectively replicate in cancer cells and stimulate an antitumor immune response. It encodes human interferon beta (IFNβ) and a recombinant CD40-ligand (MEM40) for immunotherapy against cancer [99]. An ongoing phase I clinical trial of MEM-288 (NCT05076760) consists of two parts. First is an open-label monotherapy dose escalation part that aims to determine the MTD and safety of MEM-288 in patients with various advanced cancers, including pancreatic cancer, advanced/metastatic NSCLC, cutaneous squamous-cell carcinoma (cSCC), Merkel cell, melanoma, triple-negative breast cancer (TNBC), or head and neck cancer, who have progressed following antiPD-1/PD-L1 therapy. Primary objectives include safety, tolerability, and MTD; secondary objectives focus on efficacy measures such as response rates and progression-free survival. The second part of the study is combination therapy with nivolumab, an antiPD-1 therapy. This part will evaluate the efficacy and safety of MEM-288 concurrently with nivolumab in patients with advanced/metastatic NSCLC who have relapsed after initial treatment with antiPD-1/PD-L1 with or without concurrent chemotherapy. The primary goal is to determine the overall response rate, with secondary objectives assessing safety, disease control, and other efficacy outcomes. MEM-288 is administered intratumorally every three weeks, with a maximum of six doses, while nivolumab is given intravenously at a dose of 360 mg every three weeks, with optional maintenance for up to two years [99].

VG161 is an injectable oncolytic HSV-1 therapy that incorporates recombinant human IL-12, IL-15, and PD-L1B [100]. A Chinese multicenter, open-label, phase I/II clinical trial currently recruiting has a single-arm design to assess the safety, tolerability, and preliminary effectiveness of VG161 in combination with the PD-1 inhibitor nivolumab in HSV-seropositive patients with advanced pancreatic cancer. The trial utilizes a standard 3 + 3 dose-escalation design to evaluate the safety of the combination therapy and to determine the recommended phase 2 dose (RP2D) for further efficacy testing. The first cycle of treatment is monitored up to Day 28 for dose-limiting toxicity (DLT). For the efficacy evaluation, a Simon two-stage design is implemented to further explore the preliminary effectiveness of the combination therapy at the established safe dose [100]. These clinical studies collectively illustrate the evolving landscape of OV therapies in pancreatic cancer and highlight advancements in the field that aim at improved patient outcomes. A summary of the clinical trials using OVs in pancreatic cancer therapy is presented in Table 2.

## 3. Challenges and Future Directions

OVs hold significant potential for transforming pancreatic cancer treatment, but several challenges must be addressed to fully realize their therapeutic benefits. Key considerations include viral selectivity, systemic toxicity, route of administration, host immune response, fibrotic tumor microenvironment, patient selection, and tumor heterogeneity.

A key future challenge is overcoming the obstacle posed by the hypoxic environment within solid tumors, which can hinder viral replication. Pipiya et al. studied the effects of hypoxia in lung, pancreatic, prostate, and colon cancer cells and demonstrated that while hypoxia reduced viral protein production, it did not affect mRNA levels. This underscores the need to address hypoxia-induced limitations to improve the efficacy of oncolytic therapies [101]. Widespread distribution of primary cellular receptors reduces viral selectivity, while systemic toxicity limits the dose that can be safely administered [102]. Although direct intratumoral administration is an invasive process, and thought to be less effective in treating distant metastases, the success of our trials indicates the superiority of this approach [58,103] over intravenous administration that carries risks of liver tropism, which can diminish the virus’s availability in the bloodstream and may cause systemic toxicity [102]. The host immune system poses another significant hurdle, as preexisting or treatment-induced antiviral antibodies, along with complement activation, antiviral cytokines, and macrophages, can neutralize OVs before they reach the tumor [104,105,106]. Although OVs can enhance local oncolysis, the strong immune response they trigger also raises the risk of systemic reactions which may reduce the overall efficiency and safety of the therapy [107,108]. The dense fibrotic tissue characteristic of pancreatic cancer further complicates treatment by hindering the penetration and spread of both OVs and traditional chemotherapies [102]. Components of the ECM such as heparan sulfate and collagen, along with fibrosis, necrosis, and interstitial hydrostatic pressure, create physical barriers that impede the diffusion of OVs to tumor sites [109,110]. Addressing these barriers by modifying the ECM is crucial for improving OV delivery and effectiveness. Patient selection and tumor heterogeneity add another layer of complexity as not all patients respond positively to OV therapy. The variability in tumor types, stages, and inherent patient differences makes it challenging to determine the most suitable OV treatment [110]. Identifying clinical biomarkers is essential for predicting which patients are likely to benefit, thereby improving treatment outcomes [111,112]. Recent studies have also explored the metabolic remodeling effects of OVs. Mahalingam et al. showed that oncolytic vaccinia virus could alter the metabolic profile of tumor cells by reducing glycolysis and inducing metabolic stress, leading to increased immunogenic cell death. By impairing energy production pathways, OVs not only target cancer cells directly but also make the tumor more susceptible to immune-mediated destruction [113]. Thus, exploring OV-mediated metabolic remodeling offers a promising avenue for enhancing OV efficacy and combining it with other therapies, such as metabolic inhibitors or immunotherapies.

Technological advancements, including optimizing viral delivery to tumor sites [114] and combining OVs with targeted nanoparticles [115,116,117], will reduce immune clearance and further enhance therapeutic efficacy. Promising results from multiple clinical trials that combined OVs with chemotherapy, radiotherapy, immune-modulatory and/or immune checkpoint inhibitors (Table 2) [118,119] are poised to become a crucial component for clinical management in advanced pancreatic cancer. Moreover, integrating artificial intelligence (AI) and machine learning (ML) into early screening and detection will help with personalized medicine and significantly advance OV therapy [120,121].

## 4. Conclusions

OVs represent a promising approach to cancer immunotherapy, with the potential to overcome some of the limitations associated with traditional cancer treatments. By selectively targeting cancer cells and stimulating antitumor immune responses, OVs offer a novel and versatile strategy for treating hard-to-manage solid cancers, including advanced pancreatic cancer. Continued research and clinical development are essential to realize the full therapeutic potential of OVs in cancer management.

## Figures and Tables

**Figure 1 ijms-25-09912-f001:**
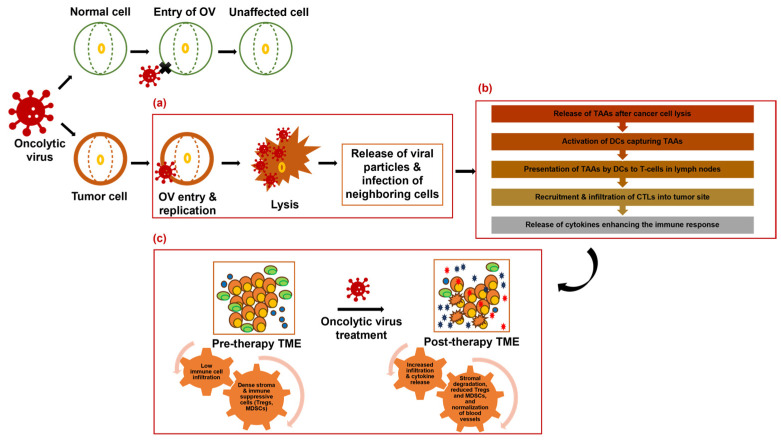
Mechanism of oncolytic virus activity and tumor microenvironment remodeling in pancreatic cancer. (**a**) OVs specifically infect and lyse cancer cells, releasing viral particles and infecting neighboring cells. (**b**) Release of tumor-associated antigens (TAAs) after cancer cell lysis leads to an immune response characterized by dendritic cell activation, T-cell priming, and recruitment of cytotoxic T lymphocytes (CTLs) to the tumor site. (**c**) OVs also remodel the tumor microenvironment (TME) by degrading stromal components, reducing immunosuppressive cells, and promoting immune cell infiltration. These combined effects result in tumor shrinkage and potentially synergize with other therapeutic modalities.

**Table 1 ijms-25-09912-t001:** Preclinical studies investigating various oncolytic viruses as potential therapies for pancreatic cancer.

Virus Name	Virus Type	Mechanism & Target	Enhancements	Efficacy in Models	Key Findings	References
Reolysin	Reovirus	Selective reovirus replication and induction of ER stress-mediated apoptosis; targets KRas-transformed cells	Combined with ER stress inducers (tunicamycin, brefeldin A, and bortezomib)	Enhanced selective replication, increased ER stress, apoptosis; tumor reduction	Effectively targets Ras-activated cancers	[71]
MV-EGFPmtd	Measles virus	Integrated MicroRNA target sites (miRTS) interact with normal tissue microRNAs, preventing viral replication in nontarget cells	--	Detargets normal tissues; retains oncolytic potency in pancreatic cancer models	Feasibility of modifying tropism without compromising efficacy	[72]
oAd/DCN/LRP-PEG-NT	Adenovirus	Modified with NT peptide and PEG for enhanced targeting, Degrades ECM, targets NTR-overexpressing cells	--	Significant tumor suppression; improved transduction efficiency	Effective targeting of NTR-overexpressing pancreatic cancer cells; ECM degradation; Wnt signaling inhibition	[62]
uMSC-delivered oAd/RLX-PCDP	Adenovirus	Delivered via hMSCs; enhances viral delivery and replication Complexed with biodegradable polymer (PCDP) for improved delivery	--	Stronger antitumor effect in pancreatic tumor models compared to naked virus or hMSC treatment	Enhanced internalization, viral production, and release in tumor tissues	[63]
VLΔTK-IL-10	Vaccinia virus	Deleted TK and armed with IL-10 genes; enhances targeted viral attack on tumors	--	Prolonged survival in immunocompetent and genetically engineered mouse models	Increased survival rates; targeted viral attack on tumors	[77]
Delta-24-RGD (DNX-2401)	Adenovirus	Selective replication in p16/RB/E2F pathway abnormal cells	Combined with phosphatidylserine targeting antibody 1N11	Reduced tumor growth; enhanced anticancer immune responses	Significant reduction in tumor growth; targeted replication in abnormal pathway cells	[61]
ICOVIR15	Adenovirus	Enhanced E1A and late viral protein expression through miR-99b and miR-485	--	Increased adenoviral activity in pancreatic cancer cell lines	Downregulation of transcriptional repressors ELF4, MDM2, and KLF8; enhanced antitumoral activity	[78]
YDC002	Adenovirus	Degrades ECM, enhances chemosensitivity	Combined with gemcitabine	Potent anticancer effects; enhanced chemosensitivity	Reduced ECM components; enhanced cytotoxicity of gemcitabine	[47]
OAds expressing IFN	Adenovirus	Expresses interferon; enhances chemotherapy efficacy	Combined with 5-FU, gemcitabine, and cisplatin	Improved cancer cell death in vitro; inhibited tumor growth in animal models	Improved efficacy of chemotherapy drugs; enhanced survival rates	[65]
SAG101	Adenovirus	Coated with OM-pBAEs for enhanced transduction	--	Improved antitumor activity and reduced toxicity in PDAC mouse models	Enhanced transduction efficiency; reduced liver sequestration; improved therapeutic potential	[64]
AdNuPARmE1A-miR222-S	Adenovirus	Engineered with miR-222 binding sites; Inhibits miR-222 to improve viral yield and cytotoxic effects in vivo	--	Controlled tumor progression better than control virus in xenografts	Enhanced viral fitness, and improved tumor control through miRNA modulation	[67]
oHSV-CD40L	Herpes simplex	Expresses membrane-bound CD40L; stimulates immune responses	Combined with PD-1 antagonist antibody	Slowed tumor growth, prolonged survival; increased mature DCs and activated cytotoxic T cells	Enhanced immune response; improved outcomes in pancreatic cancer treatment	[74]
Ad5-3Δ-A20T	Adenovirus	Selective targeting of αvβ6 integrin receptor, promoting viral propagation and spread	--	Superior efficacy in 3D organotypic cocultures and Suit-2 for in vivo models	Highly selective for αvβ6 integrin-expressing pancreatic cancer cells; potential to improve systemic delivery	[79]
OAd.R.shPKM2	Adenovirus	Knockdown of PKM2	--	Reduced tumor growth in PANC-1 xenograft model	Induced apoptosis and impaired autophagy; strong antitumor effect	[80]
ZD55-TRAIL-IETD-Smac	Adenovirus	Expression of TRAIL and Smac; Enhanced apoptosis of pancreatic cancer cells by affecting antiapoptotic signaling elements	Combined with Cyclin-dependent kinase (CDK) inhibitor SNS-032	Significant inhibition of BxPC-3 pancreatic tumor xenografts	Combination therapy sensitized cancer cells to apoptosis	[70]
OAd expressing survivin shRNA & TRAIL	Adenovirus	Downregulation of survivin with TRAIL expression to enhance cytotoxic death post-gemcitabine treatment	Combined with Gemcitabine	Tumor regression in MiaPaCa-2 pancreatic cancer model	Enhanced cell death correlated with survivin downregulation and increased PARP cleavage	[81]
VSV-ΔM51-GFP	Vesicular Stomatitis Virus	Enhance viral replication and oncolysis by targeting IKK-β and JAK1	Combined with TPCA-1 (IKK-β inhibitor) and ruxolitinib (JAK1/2 inhibitor)	Enhanced replication and oncolysis in VSV-resistant PDAC cell lines with inhibitors	Upregulated type I interferon signaling contributes to resistance; inhibition of STAT1/2 phosphorylation enhances VSV-ΔM51 efficacy	[82]
microRNA-sensitive CD-UPRT-armed MeV	Measles Virus	MicroRNA-regulated vector tropism targeting pancreatic cancer cells; 5-fluorouracil-based chemovirotherapy	Combined with 5-fluorocytosine	Delayed tumor growth and prolonged survival in xenografts with combination treatment	Effective strategy against pancreatic cancer with favorable therapeutic index; potential for clinical translation	[83]
H-1PV	Parvovirus	Induction of oxidative stress and apoptosis	Combined with Histone deacetylase inhibitors (HDACIs) valproic acid	Complete tumor remission in rat and mouse xenograft models	Promising results warranting clinical evaluation in cervical and pancreatic ductal carcinomas	[84]
GLV-1h151	Vaccinia Virus	Direct oncolysis; inflammation-mediated immune responses	Combined with radiation	Synergistic cytotoxic effect in combination with radiation; well-tolerated in mice	Potential for clinical translation; effective against colorectal cancers independent of disease stage	[85]

**Table 2 ijms-25-09912-t002:** Oncolytic virus-based clinical trials that are actively recruiting/not yet recruiting/completed in locally advanced or metastatic or relapsed pancreatic cancer.

Phase	Oncolytic Virus	Clinical Trial Identifier	Description	Inclusion Criteria	Coupled with	Virus Dose	Status	Estimated Study Completion (mm/yyyy)
I	Talimog-ene Laherpa-repvec (T-VEC)	NCT03086642	Immune-enhanced herpes simplex virus type-1 (HSV-1) that selectively replicates in solid tumors	Locally advanced or metastatic pancreatic cancer, refractory to at least one chemotherapy regimen	--	4.0 mL of 10^6^ PFU/mL on week 1 and 4.0 mL of 10^6^, 10^7^, or 10^8^ PFU/mL on week 4	Active, not recruiting	April 2026
I	TBI-1401 (HF10)	NCT03252808	Replication-competent HSV-1 Oncolytic Virus	Stage III or IV unresectable pancreatic cancer. Patients with stage IV must have failed gemcitabine-based first-line chemotherapy.	With gemcitabine + nab-paclitaxel or TS-1.	1 × 10^6^ or 1 × 10^7^ TCID50 */mL TBI-1401 (HF10) administered to the tumor in up to 2 mL	Active, not recruiting	March 2035
IIb	VCN-01	NCT05673811	Genetically modified wild-type human adenovirus serotype 5 (HAd5) with selective replication	Metastatic pancreatic cancer	Nab-paclitaxel and gemcitabine	1 × 10^13^ vp on day 1 and day 92	Recruiting	April 2025
I	R130	NCT05860374	Modified herpes simplex virus-1(HSV-1) containing the gene coding for antiCD3 scFv/CD86/PD1/HSV2-US11	Advanced solid tumors including pancreatic cancer	--	1 × 10^8^ PFU/mL, Every 7–14 days	Recruiting	March 2026
I	R130	NCT05886075	Modified herpes simplex virus-1 (HSV-1) containing the gene coding for antiCD3 scFv/CD86/PD1/HSV2-US11	Relapsed/ refractory advanced solid tumors including pancreatic cancer	--	1 × 10^8^ PFU/mL, Every 7–14 days	Recruiting	March 2025
II	H101	NCT06196671	Oncolytic adenovirus	Advanced pancreatic cancer	PD-1 inhibitor (camrelizumab)	H101 15 × 10^11^ vp on day 1	Not yet recruiting	January 2028
I	Oncolytic virus	NCT06346808	--	Preoperative therapy for patients with borderline resectable and locally advanced pancreatic cancer	AntiPD1 (camrelizumab) and chemotherapy (gemcitabine + capecitabine)	--	Not yet recruiting	May 2027
I, II	LOAd703 (delolimogene mupadenorepvec)	NCT02705196	Oncolytic adenovirus encoding TMZ-CD40L and 4-1BBL	Locally advanced pancreatic cancer	Gemcitabine + nab-paclitaxel +/− antiPD-L1 antibody atezolizumab	six doses of 5 × 1010, 1 × 1011 or 5 × 1011 vp per treatment	Recruiting	October 2025
Ib	STI-1386 (Seprehvec)	NCT05361954	Second-generation oncolytic herpes simplex virus type 1	Relapsed and refractory solid tumors including locally advanced pancreatic cancer	--	3 + 3 dose-escalation design with three dosing cohorts: 4 mL of 1 × 10^6^, 1 × 10^7^, or 1 × 10^8^/1 mL	Not yet recruiting	February 2027
I	MEM-288	NCT05076760	Conditionally replicative oncolytic adenovirus vector encoding transgenes for human interferon beta (IFNβ) and a recombinant chimeric form of CD40-ligand (MEM40)	Solid tumors including advanced/metastatic pancreatic cancer that progressed following previous antiPD-1/PD-L1 therapy with or without concurrent chemotherapy	Part 1—MEM-288 alone Part 2—With AntiPD-1 (Nivolumab)	Up to six doses of 1 × 10^10^, 3.3 × 10^10^ or 1 × 10^11^ vp given every 3 weeks	Recruiting	November 2026
I, II	VG161	NCT05162118	Recombinant human-IL12/15/PDL1B oncolytic HSV-1	Advanced pancreatic cancer	PD-1 inhibitor (Nivolumab)	1.5 × 10^8^ on day 1, 1.0 × 10^8^ on day 1 and 2 or 1.0 × 10^8^ on day 1, 2, and 3	Recruiting	December 2025
I	VCN-01	NCT02045602	Genetically modified wild-type human adenovirus serotype 5 (HAd5) with selective replication	Advanced solid tumors including pancreatic cancer	Gemcitabine and abraxane	Single intravenous injection of 1 × 10^11^, 1 × 10^12^, 3.3 × 10^12^ or 1 × 10^13^ vp	Completed	January 2020
I	VCN-01	NCT02045589	Genetically modified wild-type human adenovirus serotype 5 (HAd5) with selective replication	Advanced pancreatic cancer	Gemcitabine and abraxane	Three intratumoral administrations of 1 × 10^10^ or 1 × 10^11^ vp	Completed	September 2018
I	CAdVEC	NCT03740256	Oncolytic adenovirus expressing PD-L1 blocking antibody and IL-12	Advanced HER2-positive solid tumors including pancreatic cancer	HER2-specific CAR-T cell	Single dose of 5 × 10^9^, 1 × 10^10^, 1 × 10^11^ or 1 × 10^12^	Recruiting	December 2038
Ib	REOLYSIN (Pelareorep)	NCT02620423	Oncolytic reovirus	Advanced pancreatic adenocarcinoma	Gemcitabine, irinotecan, or leucovorin/5-fluorouracil (5-FU) with pembrolizumab	4.5 × 10^10^ TCID50 * on Days 1 and 2 of a 21	Completed	August 2018
I	OrienX010	NCT01935453	Recombinant hGM-CSF HSV-1	Solid tumors including pancreatic cancer	--	10^6^ pfu, 10^7^ pfu, 10^8^ pfu or 4 × 10^8^ pfu as single, multiple (three injections, every 2 weeks) or continuous injections every two weeks	Completed	May 2014
I	vvDD-CDSR	NCT00574977	Vaccinia virus	Solid tumors including pancreatic cancer	--	3 × 10^7^, 1 × 10^8^, 3 × 10^8^, 1 × 10^9^ or 3 × 10^9^ pfu	Completed	July 2014
II	Reolysin	NCT01280058	Reovirus	Recurrent or Metastatic Pancreatic Cancer	Carboplatin and paclitaxel	3 × 10^10^ TCID50 */day, on days 1–5 of each cycle	Completed	January 2016
I	Ad5-yCD/mutTKSR39rep-hIL12 adenovirus (Ad5-vector)	NCT03281382	Adenovirus	Metastatic pancreatic cancer	Oral 5-fluorocytosine (5-FC) and chemotherapy	1 × 10^11^, 3 × 10^11^, or 1 × 10^12^ vp	Completed	May 2019
I	Ad5-yCD/mutTK(SR39)rep-ADP (Ad5-DS)	NCT02894944	Replication-competent adenovirus-mediated double-suicide gene therapy	Locally advanced pancreatic cancer	Oral 5-fluorocytosine, valganciclovir, gemcitabine	1 × 10^11^, 3 × 10^11^, and 1 × 10^12^ viral particles/mL	Completed	April 2019
I	ONYX-015 (dl1520)	--	E1B-55kD gene-deleted replication-selective adenovirus	Locally advanced pancreatic carcinoma	--	Dose escalation from 10^8^ pfu to 10^11^ pfu	Completed	--
I	ONYX-015 (dl1520)	--	E1B-55kD gene-deleted replication-selective adenovirus	Locally advanced adenocarcinoma of the pancreas or metastatic disease with minimal/absent liver metastases	Gemcitabine (1000 mg/m^2^)	2 × 10^10^ or 2 × 10^11^ vp/treatment	Completed	--

Note: * TCID_50:_ Tissue culture infectious dose 50.

## Data Availability

Not applicable.

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
