# Peer review of "Optimizing Pancreatic Cancer Therapy: The Promise of Immune Stimulatory Oncolytic Viruses"

_ijms, 2024, doi:10.3390/ijms25189912_

Round 1

Reviewer 1 Report

Comments and Suggestions for Authors

please see the attached file

Author Response

Reviewer 1:

This review systematically summarized the mechanism of action of oncolytic virus in immune-depleted pancreatic cancer, as well as their preclinical and clinical applications. Overall, the manuscript is comprehensive, well-written, and holds certain significance as there were only a few OV-related reviews focused on pancreatic cancer in the last five years, most of which were merely focused on oncolytic adenovirus. However, there is still some work to be done to further improve the quality of this review.

We thank the reviewer #1 for their overall positive feedback on our work.

  1. Regarding the mechanism of action of the oncolytic virus, a simple illustration to show the TME-remodeling mechanism of OV is recommended as figures are easier to understand than words at first glance. An illustration is also a standard requirement for a qualified review.

We appreciate the reviewer’s suggestion to include an illustration to clarify the tumor microenvironment (TME)-remodeling by OV. We have added a figure (Figure 1) to the manuscript, which visually depicts the remodeling of the TME in pancreatic cancer. Specifically, please refer to section “c” of Figure 1, as mentioned in page 4, line 155.

  1. Regarding the clinical application of OV in pancreatic cancer, some critical updates are missing. For example, an oncolytic adenovirus H101 has recently been reported to show tumor-specific oncolysis and efficacy in pancreatic cancer-derived peritoneal metastasis with a safe profile (PMID: 38659226). This is important because peritoneal metastasis and the following malignant ascites are also an important form and complication in late-stage pancreatic cancer, which is also an unmet clinical need. Therefore, this clinical update should be added to this review.

We appreciate the input and have discussed H101 in the revised manuscript (page 13, lines 393-398) as suggested by the reviewer. Appropriate reference has been included.

  1. Regarding the preclinical exploration, an engineered oncolytic HSV was also reported to reshape the PDAC TME and the underlying mechanism was further explored using single-cell analysis (PMID: 33130310). Although this study has been mentioned in this review, the details of this study need to be added. As single-cell sequencing of different treatment groups could reveal more information to explore the mechanism of OV.

Thank you for the insightful suggestion. We have revised the manuscript to include the study on engineered oncolytic HSV, which was reported to remodel the PDAC tumor microenvironment. Additionally, we have incorporated the details of the single-cell RNA sequencing and FACS analysis from the suggested study (PMID provided by the reviewer). Please refer to page 6, lines 281-286 for the updates.

  1. Regarding future challenges, the hypoxic environment could also impose an obstacle to viral replication (PMID: 15690061), which should also be mentioned.

Thank you for your feedback. We have discussed the suggested study on the effects of hypoxia (please refer to page 19, lines 461-466).

  1. Finally, as cancer metabolism has gradually come into focus. Are there any relevant studies reporting the metabolic remodeling effect of OV? This could be a novel focus besides reshaping the stromal components and tumor vasculature.

We acknowledge the importance of including studies that report the metabolic remodeling effects of oncolytic viruses. As per reviewer’s suggestion, we have incorporated this information into the manuscript on page 20, lines 487-494.

Reviewer 2 Report

Comments and Suggestions for Authors

Manuscritpt entitled: „Optimizing pancreatic cancer therapy; the promise of immune stymulatory oncolytic viruses” is a  review article presenting the results of trials concerning implementation of oncolytic viruses in the therapy of pancreatic cancer. This paper constitutes of two main parts. First  part is devoted to  the effects of experimental studies in vitro, using pancreatic  cell lines and in vivo in animals with pancreatic cancer. The second part showed results of preclinical and clinical trials in patients with  unresectable, metastatic pancreatic adenocarcinoma.

During the last decade oncolytic viruses have been intensively  studied as a new strategy in the  treatment in this disease. The variety of  viruses have been used such as adeno-, herpes, vaccinia,  measles, parvovirus and others.  Application of oncolytic viruses improved host’s immune defense, transformed tumor microenvironment and were used as vectors carrying suicide genes for cancer cells and IL-12 in patients with metastases. Oncolytic viruses  have been used together with classic chemo- or radiotherapy resulting in therapy improvement. Viruses therapy in pancreatic cancer was presented as safe, well tolerated and increasing the survival rate in these patients. However not all patients respond positively to oncolytic viruses therapy due to  tumor heterogeneity and such treatment of patients must be highly individualized.   

Comments and suggestions:

1.      Because of many abbreviations used in the text the abbreviations lists in included at the end of paper, that is a very good idea, helpful for the reader. On my opinion it will be better  if the abbreviations were placed in alphabetical order.  Also list of abbreviations is not completed, should be  accomplished.

2.      Tables included the results of studied with oncolytic viruses both experimental (table 1) and clinical (table 2).  It should be easier to understand the mechanisms of oncolytic viruses actions (and also more attractive) if  authors show drawing or figure presenting the effects of these viruses on pancreatic cancer cells.

3.      References is are updated and properly selected but some of  needs corrections, because part of titles are written  with capital letters ( f.ex. 3, 5,12, 17, 19, 23,, 24, 29, 32, 35, 39, 40, 41, 42, 48, 54, 57, 59, 65, 66, 75, 77, 85, 87, 92, 93, 94, 102, 110, 104.). All titles should be standardized.

4.      Some references are uncompleted ( 2,3,14, 15, 19, 21, 32, 49, 56, 64, 65, 73, 78, 88, 89, 95).

Author Response

Reviewer 2:

We thank the reviewer #2 for their thorough evaluation of our manuscript and helpful suggestions that resulted in significant improvement.

  1. Because of the many abbreviations used in the text the abbreviations list is included at the end of the paper, which is a very good idea, and helpful for the reader. In my opinion, it would be better if the abbreviations were placed in alphabetical order.  Also list of abbreviations is not completed, should be accomplished.

Thank you for your feedback. We have updated the abbreviations and organized them in alphabetical order for improved clarity and ease of reference for readers. Please see pages 20-22, lines 517-580.

  1. Tables included the results of studies with oncolytic viruses both experimental (table 1) and clinical (table 2).  It should be easier to understand the mechanisms of oncolytic viruses' actions (and also more attractive) if authors show drawing or figure presenting the effects of these viruses on pancreatic cancer cells.

We appreciate the reviewer’s suggestion to include an illustration demonstrating the effects of these viruses on pancreatic cancer cells. In response, we have added Figure 1 to the manuscript, with the relevant details in section “a” on page 4, line 155.

  1. References are updated and properly selected but some of them need corrections because part of the titles is written with capital letters ( for ex. 3, 5,12, 17, 19, 23, 24, 29, 32, 35, 39, 40, 41, 42, 48, 54, 57, 59, 65, 66, 75, 77, 85, 87, 92, 93, 94, 102, 110, 104). All titles should be standardized.

We thank the reviewer for highlighting the error. We have thoroughly reviewed and corrected all references, ensuring they are now properly standardized (please refer page 22-31, line 582-973).

  1. Some references are uncompleted ( 2,3,14, 15, 19, 21, 32, 49, 56, 64, 65, 73, 78, 88, 89, 95).

We have now reviewed and corrected all references. Please refer page 22-31, line 581-972.

Reviewer 3 Report

Comments and Suggestions for Authors

This is a well written and clear review. It is easy and pleasent to read.

I would suggest to add some more information about the historical background of oncolytic viruses which is very briefly discussed.

I would also suggest to include some images illustrating the activity of oncolytic viruses and their immunological effects. I am including an example taken from Shi, T., Song, X., Wang, Y., Liu, F., & Wei, J. (2020). Combining oncolytic viruses with cancer immunotherapy: establishing a new generation of cancer treatment. Front Immunol. 2020; 11: 683. Publisher’s Note Springer Nature remains neutral with regard to jurisdictional claims in published maps and institutional affiliations.

Author Response

Reviewer 3:

This is a well-written and clear review. It is easy and pleasant to read.

We thank Reviewer #3 for their insightful suggestions, which have greatly contributed to the improvement of our manuscript.

  1. I would suggest adding some more information about the historical background of oncolytic viruses which is very briefly discussed.

As per the reviewer’s suggestion, we have included information on the historical background of oncolytic viruses in the manuscript. Please see page 2, lines 50-58 for the relevant section.

  1. I would also suggest including some images illustrating the activity of oncolytic viruses and their immunological effects. I am including an example taken from Shi, T., Song, X., Wang, Y., Liu, F., & Wei, J. (2020). Combining oncolytic viruses with cancer immunotherapy: establishing a new generation of cancer treatment. Front Immunol. 2020; 11: 683. 

We appreciate the reviewer’s suggestion to include an illustration depicting the activity of oncolytic viruses and their immunological effects. In response, we have incorporated Figure 1 into the manuscript, with the relevant details provided in section “B” on page 4, line 155.